

# Field measurements of biogenic volatile organic compounds in the atmosphere using solid-phase microextraction Arrow

Luís Miguel Feijó Barreira[1], Geoffroy Duporté[1], Tuukka Rönkkö[1], Jevgeni Parshintsev[1], Kari Hartonen[1], Lydia Schulman[1], Enna Heikkinen[1], Matti Jussila[1], Markku Kulmala[2], Marja-Liisa Riekkola[1*]

[1] Department of Chemistry, P.O. Box 55, FI-00014 University of Helsinki, Finland
[2] Department of Physics, P.O. Box 64 Matti Jussila[1], FI-00014 University of Helsinki, Finland

*Correspondence to*: Marja-Liisa Riekkola (marja-liisa.riekkola@helsinki.fi)

**Abstract.** Biogenic volatile organic compounds (BVOCs) emitted by terrestrial vegetation participate in a diversity of natural processes. These compounds impact both on short-range processes, such as on plant protection and communication, and on

high-range processes, by e.g. participation on aerosol particle formation and growth. The biodiversity of plant species around the Earth, the vast assortment of emitted BVOCs, and their trace atmospheric concentrations contribute to the high remaining uncertainties about the effects of these compounds on atmospheric chemistry and physics, and call for the development of novel collection devices that can offer portability with improved selectivity and capacity. In this study, a novel solid-phase microextraction (SPME) Arrow sampling system was used for the static and dynamic collection of BVOCs from the boreal

forest, and samples were subsequently analysed on-site by gas chromatography-mass spectrometry (GC-MS). This system offers higher sampling capacity and improved robustness than the traditional equilibrium-based SPME techniques, such as SPME fibers. Field measurements were performed in summer 2017 at the Station for Measuring Ecosystem-Atmosphere Relations (SMEAR II) in Hyytiälä, Finland. Complementary laboratory tests were also performed to compare the SPME-based techniques under controlled experimental conditions and to evaluate the effect of temperature and relative humidity on their

extraction performance. The most abundant monoterpenes and aldehydes were successfully collected. A significant improvement on sampling capacity was observed with the new SPME Arrow system when compared to SPME fibers, with collected amounts being approximately 2 times higher for monoterpenes and 7-8 times higher for aldehydes. BVOC species exhibited different affinities for the type of sorbent materials used (PDMS/Carbon WR vs. PDMS/DVB). Higher extraction efficiencies were obtained with dynamic collection prior to equilibrium regime, but this benefit during the field measurements

was small probably due to the natural agitation provided by the wind. An increase in temperature and relative humidity caused a decrease in the amounts of analytes extracted under controlled experimental conditions, even though the effect was more significant for PDMS/Carbon WR than for PDMS/DVB. Overall, results demonstrated the benefits and challenges of using SPME Arrow for the sampling of BVOCs in the atmosphere.



## 1 Introduction

Vegetation covering Earth landmasses release a diversity of biogenic volatile organic compounds (BVOCs), which compromise a large variety of molecules that differ in size, physicochemical properties and metabolic origin (Laothawornkitkul et al., 2009; Peñuelas and Llusià, 2001; Peñuelas and Staudt, 2010). Terrestrial biosphere function as one of the key regulators of atmospheric chemistry, and is fundamental for sustainability of air quality and climate (Arneth et al., 2010; Bryan and Steiner, 2013). BVOCs participate in many natural processes, including plant metabolism, growth, reproduction, protection, and communication between plants, within plant communities, and between plants and insects (Laothawornkitkul et al., 2009; Peñuelas and Staudt, 2010). These emissions vary considerably in time, space and between species, and are strongly influenced by temperature and light (Kesselmeier and Staudt, 1999; Schollert et al., 2014; Tarvainen et al., 2005). Once in atmosphere, BVOCs participate in atmospheric reactions, which leads to the formation of numerous secondary products (Atkinson and Arey, 2003). The lifetime of BVOCs vary to a large extend, depending on the compound and oxidants involved (Atkinson and Arey, 2003). The low volatile secondary products formed in these photo-oxidation reactions can subsequently lead to the formation of secondary organic aerosols (SOA) (Jimenez et al., 2009). Aerosols are recognized to affect climate, both directly by reflecting or absorbing solar radiation and indirectly by acting as a cloud condensation nuclei (Kulmala et al., 2004). BVOCs are believed to be the largest source of SOA on a global scale (Henze and Seinfeld, 2006).

Monoterpenes are a class of naturally occurring compounds with great importance to atmospheric physics and chemistry. These compounds participate in photochemical reactions that affect ozone and carbon monoxide concentrations, and contribute to secondary organic aerosol formation and growth through their oxidation products (Kavouras et al., 1999). They also have an important biological role, e.g. as allopathic and defense compounds against pathogens and herbivores (Kesselmeier and Staudt, 1999). Measurement of monoterpenes is usually performed by on-line proton transfer reaction mass spectrometer (PTR-MS) (e.g. Aalto et al., 2014; Rantala et al., 2015). This technique offers fast detection of VOCs, good sensitivity, good time resolution and low detection limits (Graus et al., 2010). However, PTR-MS cannot differentiate compounds with the same molecular mass and it usually requires the use of long sampling lines that cause sample alteration. Alternatively, monoterpenes can be sampled on tubes packed with an adsorbent material (such as Tenax TA/Carbopack-B), and subsequently desorbed into a thermodesorption gas chromatograph-mass spectrometer for off-line or on-line analysis (e.g. Haapanala et al., 2012; Hakola et al., 2012; Wong et al., 2013). However, this technique is laborious, expensive, prones to contamination and requires additional resources/installations (e.g. cryofocusing). Alternatively, solid-phase microextraction (SPME) can be used for monoterpene sampling (e.g. Yassaa et al., 2010; Zini et al., 2001). This technique combines sampling and pre-concentration of analytes in a single step and allows for direct thermodesorption into a heated gas chromatograph injection port (Koziel et al., 1999).

Carbonyl compounds also play an important role in the atmosphere due to their involvement in photochemical reactions and contribution to aerosol particle formation and growth (Jang and Kamens, 2001; Kesselmeier and Staudt, 1999). Aldehydes have been sampled in forest air with cartridges coated with 2,4-dinitrophenyl hydrazine (DNPH) derivatization reagent and



analyzed by liquid chromatography-mass spectrometry (LC-MS) (e.g. Hellén et al., 2004). The main drawbacks of this method are the laborious sample preparation and long sampling times.

The trace amounts of BVOCs found in ambient air and their wide variety demands for the development and application of more selective and robust sampling and pre-concentration techniques. In our previous research, solid-phase microextraction

(SPME) fibers and needle trap microextraction (NTME) syringes, combined with portable GC-MS, have been successfully used for the sampling of BVOCs in the boreal forest (Barreira et al., 2015; Barreira et al., 2016). These methods have several advantages, such as full portability, low infrastructure/resource demands, high pre-concentration, no sample preparation and fast on-site analysis. However, the low mixing ratios of some of these compounds in forest atmosphere claim for additional improvements in pre-concentration. In this study, SPME Arrow was tested for the collection of BVOCs from boreal forest

ambient air. This novel SPME-based system consists of a steel rod coated with a larger amount of sorbent material than the traditional SPME fibers, offering increased capacity but maintaining the compatibility for desorption and analysis in a conventional GC-MS due to its shape and dimensions (Helin et al., 2015). The coated rod can be withdrawn in a steel tube, which makes the device more robust. Samples were collected simultaneously by SPME fibers for comparison purposes. The effect of meteorological parameters at the sampling place on the measured amounts of BVOCs was evaluated. The inherent

characteristics of SPME-based sampling techniques and coating materials used in this work (polydimethylsiloxane/divinylbenzene (PDMS/DVB) and PDMS/Carbon WR) were studied in the laboratory before the field campaign. Also, static and dynamic SPME Arrow collection modes were compared.

## 2 Material and Methods

### 2.1 Chemicals and materials

α-Pinene (98%), $\Delta^3$-Carene (≥98.5%), Limonene (≥99%), octanal (99%), nonanal (98%), decanal (≥98%) from Sigma-Aldrich (St. Louis, USA) were used as standards. Stock solutions were prepared in dichloromethane (99.99%, Fisher Scientific, Loughborough, UK), and subsequently diluted with the same solvent to obtain needed concentrations. These solutions were used for the calibration of instrument response. For laboratory studies, diffusion vials were prepared by adding small amounts of standards to headspace vials (20 mL) and inserting a piece of deactivated fused silica retention gap (1.5 m × 0.53 mm (i.d.),

Agilent Technologies, Palo Alto, USA) through the septa, to allow a constant diffusion of the compounds from the vials. These vials were inserted into a home-made permeation oven, the calibration gas flow was diluted with nitrogen, and subsequently transferred to an additional chamber from where samples were collected. The diffusion rates were measured by weighting the vials in five different days and determining the amount of analyte losses per time. The obtained diffusion rates were 0.252 mg/h for α-Pinene, 0.129 mg/h for $\Delta^3$-carene and 0.070 mg/h for octanal, which correspond to concentrations of 149 ppbv for

α-pinene, 76 ppbv for $\Delta^3$-carene and 44 ppbv for octanal. Decanal diffusion rate was not possible to determine, probably due to slow evaporation from the diffusion vial. SPME fibers coated with PDMS/DVB (65 μm, Supelco, Bellafonte, PA, USA) and PDMS/Carbon WR (95 μm, CTC Analytics AG, Zwingen, Switzerland), and SPME Arrows coated with the same





PDMS/Carbon WR (120 µm) and PDMS/DVB (120 µm) type of sorbents (CTC Analytics AG, Zwingen, Switzerland) were used for analyte collection. The SPME fibers had a sorbent length of 10 mm, while SPME Arrows sorbent length was 20 mm. The diameter of Arrow needle was 1.1 mm. All SPME fibers and SPME Arrows were pre-conditioned according to the manufacturer´s instructions.

## 2.2 Measurement site

 BVOC sampling was performed at the SMEAR II station (Station For Measuring Ecosystem-Atmosphere Relations, 61°50.845′ N, 24°17.686′ E, 179 m above sea level) in Hyytiälä, Southern Finland (Hari and Kulmala, 2005). The station is situated in an approximately 55 years old and relatively homogeneous Scots pine stand, of about 21 m canopy height and 1170 steams ha$^{-1}$ of average tree density. The forest around the station is dominated by conifers (mainly Scots pine and Norway spruce). Tampere is the largest neighboring region, with approximately half-million inhabitants, and is located 60 km southwest from the SMEAR II station. The sampling site was situated about 1 m from a 127 m high mast for atmospheric and flux measurements mounted 2 m above the average forest floor.

## 2.3 Sampling and analysis

 Ambient air samples were collected and analysed on-site from 11[th] to 15[th] of August, 2017. For comparison purposes, one PDMS/DVB SPME fiber, two PDMS/DVB SPME Arrows and one PDMS/Carbon WR SPME Arrow were used. Samples were collected in static mode for 45 minutes. Additionally, a home-made dynamic sampling system for SPME Arrow was used for comparison with static SPME collection (Fig. S1). This device was adapted and modified from the sampling system developed in previous research for SPME fibers (Barreira et al., 2015). Samples were measured using a conventional GC-MS, consisting of an Agilent 6890 N gas chromatograph equipped with an Agilent 5973 mass selective detector (Agilent Technologies, Palo Alto, USA). The analytical column was a HP-5MS (30 m × 0.25 mm x 0.25 µm, Agilent Technologies, Palo Alto, CA, USA). The initial oven temperature was 70°C (1 min), and it was increased to 250°C (1 min) at 20°C/min. The total run time was 11 minutes. Helium (99.996%, AGA, Espoo, Finland) was used as carrier gas in a constant flow mode (1.5 mL/min). SPME Arrow and fibers were desorbed in splitless mode (2 min) with a 2.0 mm internal diameter (i.d.) split/splitless inlet liner. Desorption temperature was 270°C for all the SPME devices. A standard inlet septum was used for SPME Arrow, while a 23-gauge Merlin Microseal and a Merlin nut (Merlin Instrument Company, Half Moon Bay, USA) were used in the injection port for conventional SPME fibers. The temperature of GC-MS transfer line was 250°C and the ion source and quadrupole temperatures were kept at 230°C and 150°C, respectively. Electron ionization (70 eV) was used. The scan mass range was from 30–400 amu. The mass spectra and retention times of each analyte were obtained with standard solutions and used for identification of studied compounds in the collected samples. For semi-quantitation, extracted ion chromatograms with base ions were used (m/z 93 for α-pinene, $\Delta^3$-carene and limonene; m/z 43 for octanal and decanal; and m/z 57 for nonanal).



The same method was employed for the laboratory tests, although the initial oven temperature was 50°C (1 min) and the final temperature 250°C (1 min) at 20°C/min. For the laboratory determination of analytes extraction time profiles and to compare the extraction efficiencies of the different SPME-based sampling techniques, an Agilent 5975 C mass selective detector (Agilent Technologies, Palo Alto, USA) was also used, while all other laboratory tests were performed with the same GC-MS

used for the field measurements. Samples were collected during 10 minutes, except for studying the effects of temperature and relative humidity on the collection of BVOCs in which a 20 minutes sampling time was chosen.

# 3 Results

In this work, the SPME Arrow system was optimized and tested in the laboratory to study its applicability for the field measurement of BVOCs in forest atmosphere. The characterization of SPME-based techniques, including kinetics of

extraction, comparison of techniques and adsorbents extraction performances, and influence of temperature and relative humidity on the extracted amounts are described in the first sections. The measurements performed in the field and comparison with atmospheric temperature, relative humidity, ozone (measured at 4.2 m height), precipitation, photosynthetically active radiation (PAR) and particle number concentration (PNC) (available at http://avaa.tdata.fi/web/smart and provided by Junninen et al., 2009) are then further discussed.

## 3.1 Characterization of SPME-based sampling techniques

### 3.1.1 Extraction time profile

The extraction time profiles for α-pinene, $\Delta^3$-carene, octanal and decanal were obtained in this study, to evaluate the occurrence of equilibrium and/or competitive adsorption during an experimental sampling time of one hour at laboratory ambient temperature. These compounds have been reported as the most abundant monoterpenes and aldehydes at the sampling site (e.g.

Barreira et al., 2016). α-Pinene reached equilibrium after 10 minutes of extraction when a SPME Arrow coated with PDMS/DVB was used, and after 20 minutes with the SPME fiber coated with the same material (Fig. S2). For PDMS/Carbon WR, equilibrium was reached already after 5 minutes of sampling for both SPME-based techniques. $\Delta^3$-Carene did not reach equilibrium when using PDMS/DVB for one hour, while with PDMS/Carbon WR equilibrium was reached after 20 minutes for SPME Arrow and 40 minutes for SPME fiber. These results show that kinetics of extraction are faster with PDMS/Carbon

WR comparatively to PDMS/DVB. The extraction time profiles for the aliphatic aldehydes studied showed that these compounds did not reach equilibrium after 60 minutes of sampling.

In the obtained extraction profiles, analyte amounts did not decrease with time when using a SPME fiber and SPME Arrow coated with PDMS/DVB and a SPME fiber coated with PDMS/Carbon WR. This evidence suggests that interanalyte displacement due to competitive adsorption  was not observed during the period of sampling, which is particularly interesting

for PDMS/DVB since the uniformity of its micropores has been reported to potentially result in the displacement of analytes



with less affinity by the ones with highest (Pawliszyn, 2011). However, when using SPME Arrow coated with PDMS/Carbon WR, some displacement was observed for $\Delta^3$-carene.

The extraction time profile for a dynamic sampling with a SPME Arrow coated with PDMS/Carbon WR was also obtained (Fig. S3). As expected, equilibrium was reached much faster than in static mode. This fact is clearly observed for octanal, that

reached equilibrium in 40 minutes while in static mode it was not achieved during the experimental time. However, displacement effects were as well observed for $\Delta^3$-Carene.

### 3.1.2 Extraction efficiencies of different SPME-based techniques

A comparison between SPME Arrow and SPME fiber extraction efficiencies was performed under the laboratory conditions described in the previous section. As shown in Fig. S4, extraction efficiency of SPME Arrow was approximately 2 times higher

than for SPME fiber for PDMS/DVB and 3 times higher for PDMS/Carbon WR after 10 minutes of sampling, even though it was slightly different depending on the analytes. Compound specific extraction efficiencies were distinct for PDMS/DVB and PDMS/Carbon WR. PDMS/DVB had the best affinity towards monoterpenes, while there was no statistically significant difference between materials for aldehydes. Both materials adsorbed more $\Delta^3$-carene than α-pinene. At first glance, this selective adsorption seems to be greater for PDMS/Carbon WR than for PDMS/DVB. However, the equilibrium was not

reached for all the studied compounds and a longer sampling time, such as the one used in the field measurements (45 min), will then impact on the relative amounts of analytes collected with both materials (Fig. S3). Extraction efficiencies of sampling modes were also compared. As observed in Fig S5, α-pinene amounts were similar with both static and dynamic sampling. This result is expected since when equilibrium is reached the analyte collection is not anymore influenced by the sampling mode used. However, kinetics of extraction are much faster with dynamic sampling, which results in higher amount of analyte

extracted in shorter time. On the other hand, the collected amounts of $\Delta^3$-carene and studied aldehydes were higher with dynamic mode under pre-equilibrium conditions.

### 3.1.3 Effect of temperature and relative humidity on the extraction

The effect of temperature and relative humidity on the SPME Arrow extraction efficiencies for the analytes were studied under controlled conditions. The total sampling time was 20 minutes, which was considered to be enough to note any effect of this

parameters on the extracted amounts. An increase in temperature is recognized to affect SPME collection by decreasing the distribution constant. The change in distribution constant with temperature is also dependent on the molar change in enthalpy of the analyte when it moves from the gas phase to the fiber sorbent (Pawliszyn, 2011). According to our results, the temperature effect changes significantly depending on the coating material and the analyte. As observed in Fig. 1, the effect of temperature was more significant for α-pinene extraction, especially when using a PDMS/Carbon WR material. On the

other hand, temperature had a smaller effect on the extraction of $\Delta^3$-carene, also more pronounced when using a PDMS/Carbon WR adsorbent. Extraction efficiency of octanal and decanal was not influenced by temperature. Similar results were obtained for SPME fibers coated with the same sorbents (Fig. S6). These results are expected due to the differences in molar change in



enthalpy for the analytes, which causes distinct changes in the partition coefficient at different temperatures. An underestimation of α-pinene measured amounts is then expected when semi-quantifying monoterpenes under experimental conditions where temperature changes are quite significant.

The effect of relative humidity on extraction efficiency of SPME fibers has been observed previously (e.g. Namieśnik et al., 2003). However, in our study, this effect was expected to be small due to the high hydrophobicity of SPME coating materials used. Indeed, relative humidity had a small influence on the extraction performances of both SPME Arrows (Fig. 2) and SPME fibers (Fig. S7). The small effect of relative humidity when using hydrophobic materials has been also observed previously, where the extraction of benzene, toluene, p-xylene and ethylbenzene with a PDMS/DVB coated SPME fiber at different humidity showed a maximum reduction of the mass adsorbed by approximately 21% after 1-h sampling (Koziel et al., 2000). A small effect of relative humidity on the SPME extraction when using hydrophobic coatings was also shown in another study, where two carbon-based SPME coatings were used for the sampling of 1,1,1,-trichloroethane and carbon tetrachloride (Chai and Pawliszyn, 1995). In the referred publication, relative humidity reduced the amounts of extracted analytes at ambient temperature by less than 10% at up to 75% RH. Longer sampling times will probably enhance the effects of both temperature and relative humidity on the extracted amounts of analytes, but additional studies are still required to verify this hypothesis.

### 3.2 Calibration of instrument response

The calibration of instrument response was performed for monoterpenes (α-pinene, $\Delta^3$-carene and limonene) and aldehydes (octanal, nonanal and decanal), to estimate the mass adsorbed on the coating materials of the different SPME-based devices (Table S1). A linear 4-point calibration curve (0.1 ng to 10 ng) was obtained for monoterpenes, while a linear 5-point calibration curve (0.1 ng to 50 ng) was obtained for aldehydes. Three repetitions were done for each concentration level. The intermediate reproducibility (Rw), expressed as relative standard deviation (RSD), was from 0.2 to 19.5% for monoterpenes and from 2.0 to 18.7% for aldehydes, with higher RSD for lower concentration levels. A good linearity and sufficient correlation coefficients were observed for the mass ranges used. The limits of detection (LOD), which are also given in Table S1, varied from 17.7 pg to 28 pg for monoterpenes, while the LOD values obtained for aldehydes were from 61.1 pg to 155.2 pg.

### 3.3 Atmospheric levels of organic volatile compounds identified in air samples

In this study, three monoterpenes, namely α-pinene, $\Delta^3$-carene and limonene, were identified and measured (Fig. S8 and Table S2). The extracted amounts of BVOCs were in the order of few ng, which are in line with our preliminary measurements done in the previous year using an SPME Arrow coated with PMDS/Carbon WR and the same method as in this work (Fig. S9). The atmospheric levels of these BVOCs have been intensively determined at the SMEAR II boreal forest station, and are known to be dominated by α-pinene and $\Delta^3$-carene (Rinne et al., 2000; Yassaa et al., 2012). Similar results have been as well found in our previous research by using a SPME fiber (PDMS/DVB) combined with portable GC-MS (Barreira et al., 2015). Limonene has been also found in previous research, but at relative amounts that were considerably smaller when compared to





the most abundant monoterpenes (e.g. Barreira et al., 2015; Rinne et al., 2000; Yassaa et al., 2012). The dominance of α-pinene and $\Delta^3$-carene was observed in this study, when considering the peak areas obtained by conventional GC-MS (Fig. S10). However, when the calibration of analytes response with standard solutions was performed (but not the calibration of SPME collection), $\Delta^3$-carene and limonene levels increase relatively to α-pinene (Fig. 3A). This fact is particularly significant for

limonene, which even overcomes the levels of α-pinene when SPME ARROW is used as a sampling device. Furthermore, PDMS/Carbon WR enhanced these differences when compared to PDMS/DVB. These results show that the materials used in this study are particularly selective for these compounds, especially for limonene. The preferential adsorption of both materials for $\Delta^3$-carene comparatively to α-pinene was also demonstrated in the laboratory studies described earlier (section 3.1.2). Furthermore, $\Delta^3$-carene amounts are enhanced by the temperature and relative humidity effects on α-pinene extraction

described in section 3.1.3, this effect being more marked for PDMS/Carbon WR.

Aliphatic aldehydes, particularly octanal, nonanal and decanal, were identified in this study (Fig. S8 and Table S2). These aldehydes have been as well reported in another research performed in the boreal forest (Hellén et al., 2004). The most abundant aldehydes measured during the sampling campaign were nonanal and decanal, while octanal amounts were relatively small and only measurable when a SPME Arrow device was used (Fig. 3B). These aldehydes were more adsorbed with PDMS/DVB

than with PDMS/Carbon WR, which was not observed in the laboratory experiments. The reason for this result might be related with the different times of collection and kinetics of adsorption, since equilibrium has not been reached in those experiments. During field measurements, the presence of wind speed or air bulk movement significantly affects the mass transfer process from bulk air to the sorbent (Pawliszyn, 2011), which can reveal differences in the adsorption of these compounds.

A comparison between SPME fiber and SPME Arrow was also performed, to evaluate quantitatively the enrichment provided

by the SPME Arrows. Only PDMS/DVB was used for comparison in this study, since it was the less sensitive material to changes in temperature and relative humidity during the laboratory experiments. The monoterpenes were highly enriched when using SPME Arrow instead of the conventional SPME fiber (Fig. 4A). The amounts of monoterpenes measured when using SPME Arrow were approximately 2 times more than with the SPME fiber of the same material, which resembles the results obtained in the laboratory experiments. The enhancement was also slightly different depending on the analytes. However,

compound specific extraction efficiencies were observed between these devices. The ratio between $\Delta^3$-carene and α-pinene was approximately 1.4 when using an SPME fiber, but raised to 1.8 with SPME ARROW. This fact was found as well in the laboratory experiments, where a ratio of 0.9 was obtained for these compounds with the SPME fiber while a ratio of 1.3 was observed for SPME Arrow. The difference is less significant between limonene and pinene, with a ratio of 1.0 for the SPME fiber and 1.1 for the SPME ARROW; and between $\Delta^3$-carene and limonene, with ratios of 1.5 and 1.6 respectively.

The SPME Arrow system also improved the collection of the aliphatic aldehydes studied when compared to SPME fibers (Fig. 4B). Interestingly, the improvement effect of SPME ARROW was much higher than the one verified in the laboratory studies, with amounts increasing 7 to 8 times comparatively to the SPME fiber. However, the sampling time for the laboratory experiments was shorter and equilibrium has not been reached for aldehydes. Furthermore, the wind influences the mass transfer process from air to the sorbent during the field experiments. Both of these facts impact on the amounts adsorbed in



the SPME devices and can cause the observed differences in the enrichment with SPME Arrow. The compound specific extraction was also evaluated in this study for nonanal and decanal, since octanal has not been detected with SPME fiber. Accordingly, the ratio between decanal and nonanal was 2.8 with SPME fiber and 2.4 with SPME Arrow, which indicates a small increase in nonanal extraction when using SPME Arrow. In the laboratory experiments, the ratio decreased from 0.18 to

0.13, which corroborates the field results.

Static and dynamic collection were compared for SPME ARROW coated with PDMS/DVB. The extraction amounts were slightly higher when dynamic sampling was used. This result was observed for the all monoterpenes (Fig. 5A) and aldehydes (Fig. 5B) identified and measured during the sampling campaign. However, the differences were relatively small. This result suggests the proximity to or the attainment of the equilibrium state, where an increase in the time of extraction does not result

in higher amounts of analyte extracted on the SPME materials. However, when collection is performed in dynamic mode, compounds with lower affinity for the coating are more susceptible to displacement effects (e.g. Tuduri et al., 2002). A decrease on extraction time can partially prevent this effects, even though might also cause a decrease in sensitivity that can compromise the possibility to measure some of the BVOCs present at trace levels in the atmosphere. The effect of wind speed cannot also be neglected when using these devices, since it will increase the flow during the sampling in a similar way to the sampling

devices used for dynamic extraction. Besides, VOC mass loading on the sorbent increases with an increase in wind velocity from 0 to 5 cm/s (Pawliszyn, 2011).

### 3.4 Effect of meteorological parameters on the atmospheric levels of VOCs

The effect of meteorological parameters (Table S3) on the measured atmospheric levels of monoterpenes was evaluated in this study, in order to understand their influence on the amounts adsorbed on the coating materials of the SPME-based sampling

devices used (Fig. 6). The ratio between the amounts of monoterpenes and aldehydes sampled with PDMS/Carbon WR and PDMS/DVB was also compared with the referred parameters, in order to understand if meteorological conditions affect differently these two different types of extraction materials (Fig. S.11). Extraction temperature has two opposing effects in field measurements performed at boreal forest sites, since increasing the temperature enhances VOC emissions from Scots pine, but because adsorption is an exothermic process, increasing temperature will reduce the distribution constant of the

analytes (de Fatima Alpendurada, 2000; Tarvainen et al., 2005). However, the temperature remained almost constant during the campaign, with a variation from 15 to 20 °C. Due to this reason, the effect of temperature on the measured amounts is expected to be small when compared to other parameters influencing the analyte emissions and atmospheric concentrations. This small effect was found in our study, where any significant correlation was found between temperature and the adsorbed amounts of monoterpenes. However, some anti-correlation was found for the ratio between the amounts sampled with

PDMS/Carbon WR and PDMS/DVB, which is explained by the fact that PDMS/Carbon WR is more affected by changes in temperature than PDMS/DVB (section 3.1.3).

Relative humidity (RH) and precipitation also have two opposition effects, since monoterpene emission rates increase at high humidity levels and during and after precipitation (Llusià and Peñuelas, 1999; Schade et al., 1999), but also causes a small



decrease in the SPME extraction capacity (section 3.1.3). As can be verified in Fig. 6, some correlation was observed when relative humidity was high. This correlation is more noticeable during and after rain episodes. However, the ratio between the amounts of monoterpenes sampled with PDMS/Carbon WR and PDMS/DVB increased at low humidity and decreases when humidity is high. This result also reflects the more pronounced effect of relative humidity on PDMS/Carbon WR material.

Due to the referred constancy of temperature during the sampling campaign, ozone and PAR were expected to be the most significant meteorological parameters affecting the measured monoterpene amounts. Indeed, an anti-correlation was found between the measured monoterpenes and these parameters. However, both ozone and PAR did not affect distinctively the adsorption on the two different materials used in this study, which demonstrates that they impact is mostly on the monoterpene mixing rations in the atmosphere. Nonetheless, some oxidation might occur after collection on the SPME devices, even though

the total collection time was inferior to the lifetime of the studied compounds when exposed to ozone (Atkinson and Arey, 2003). PNC increased with the amount of monoterpenes present in the ambient air. This result is expected, since monoterpene oxidation in the atmosphere and consequent formation of low volatile compounds have been recognized to contribute to aerosol particle formation and growth (Laaksonen et al., 2008). Additionally, low amounts of monoterpenes were found in days when particle number concentration was extremely high. This result is also indicative of the oxidation of monoterpenes in the

atmosphere and partition into aerosol particle phase, under favorable ambient conditions.

The effect of meteorological parameters on the measured amounts of aliphatic aldehydes was also studied (Fig. 7). The ratio between the amounts sampled with PDMS/Carbon WR and PDMS/DVB was as well used for comparison (Fig. S11). The atmospheric levels of these compounds seemed to increase with ambient temperature. Some correlation of carbonyl compounds with temperature has been described by Hellén et al. (2004). However, as referred previously, temperature remained almost

constant during all sampling campaign. Furthermore, studies about the temperature effect on carbonyl emissions from forest are still scarce. On the other hand, some negative effect of relative humidity and precipitation on the amounts of aldehydes was evident during the campaign period, even though the effect is not visible when concentrations are high. This result might be explained by the fact that these compounds dissolve in water at low concentrations. However, also more studies under controlled conditions are required to demonstrate this hypothesis. A correlation was also found between aldehydes and ozone,

which is expected since ozone is known to increase aldehyde emissions from vegetation (Wildt et al., 2003). On the other hand, any correlation was observed between PAR or PNC and the measured amounts of aldehydes. This result is explained by the lower reactivity of aldehydes in the atmosphere when compared to terpenoid compounds (Hellén et al., 2004). The studied aldehydes might yet contribute to aerosol formation and growth, but any inference about these contribution is very challenging without additional studies performed under controlled conditions. The ratio between the amounts of aldehydes sampled with

PDMS/Carbon WR and PDMS/DVB was not affected by any of meteorological parameters (Fig. S11).

Even though the measured amounts of studied BVOCs followed the meteorological parameters registered at the sampling site, the amounts of the analytes extracted did not vary markedly during the sampling campaign to really obtain a proof for the correlation. This result is logical since temperature, which also did not vary significantly, is the main driving force of monoterpene emissions during summer and also influences aldehyde emissions (Tarvainen et al., 2005; Wildt et al., 2003). An




exception was verified in the second day of campaign, where a peak on both monoterpene and aldehyde concentrations was observed. This peak coincided with a period of high temperature, high humidity, precipitation and the lowest PAR registered during the sampling periods. Since a similar trend in concentrations was observed for monoterpenes and aldehydes, this finding might indicate that studied aldehydes respond predominantly to the same atmospheric conditions contributing to monoterpene emissions. However, additional studies are still required to confirm this possibility.

## 4 Conclusion

A novel SPME Arrow system was tested in this study for the collection of BVOCs at the boreal forest (SMEAR II, Hyytiälä, Finland). Conventional SPME fibers were used for comparison. Samples were successfully collected with both SPME-based sampling systems, and were analyzed by conventional GC-MS. Neither additional sampling line nor sample pre-treatment was needed, reducing analysis time, sample contamination and potential losses. The most abundant monoterpenes and aldehydes were measured. PDMS/Carbon WR had higher affinity towards $\Delta^3$-carene and limonene than PDMS/DVB, while PDMS/DVB enhanced the extraction of α-pinene. Nonanal and decanal were the most abundant aliphatic aldehydes. The extraction efficiency of the SPME Arrow was about two times higher than the one of SPME fiber, with an exception for aldehydes during the field campaign where a 7 to 8-fold enhancement was observed. Dynamic sampling demonstrated higher extraction efficiencies than static mode prior to equilibrium, but the improvement during field measurements was small due to the effect of wind speed on the extraction and/or to the fact that extraction was near equilibrium. Meteorological parameters influenced the amounts of studied BVOCs in the atmosphere. Laboratory tests showed that temperature and relative humidity decrease the extracted amounts of BVOCs, especially the ones with higher volatility and when PDMS/Carbon WR is used. Overall, results demonstrated the potential of SPME Arrow for the in-situ measurement of BVOCs in the atmosphere and the challenges that need to be solved for using these devices for quantitative purposes. More studies under controlled conditions are needed to understand the influence of co-adsorbed species and to develop a proper calibration method for field measurements.

## Acknowledgements

Financial support was provided by the Academy of Finland Center of Excellence programme (grant no. 307331). CTC Analytics AG (Zwingen, Switzerland) and BGB Analytik AG (Zurich, Switzerland) are thanked for the cooperation. The staff of Laboratory of Analytical Chemistry and Smear II station are thanked for the collaboration.

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




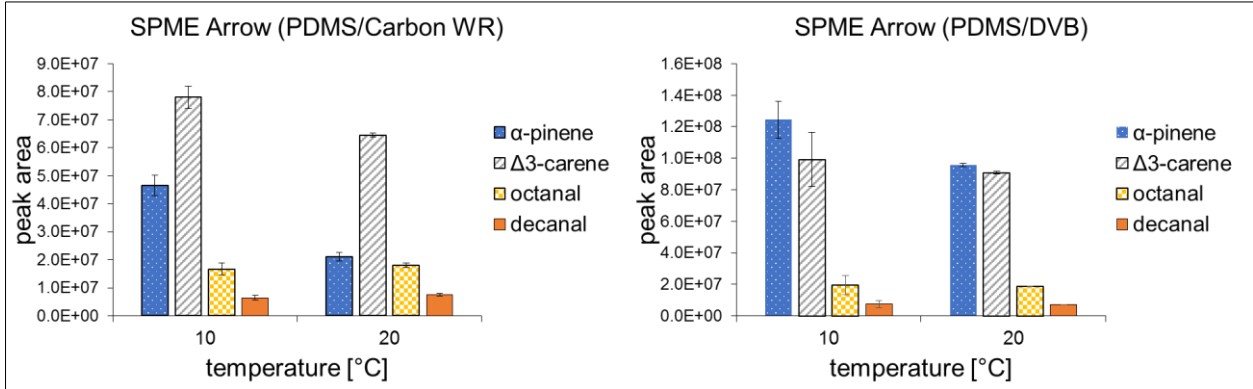

**Figure 1: Effect of temperature (°C) on the extraction efficiencies obtained with SPME ARROW using PDMS/Carbon WR and PDMS/DVB sorbents.**

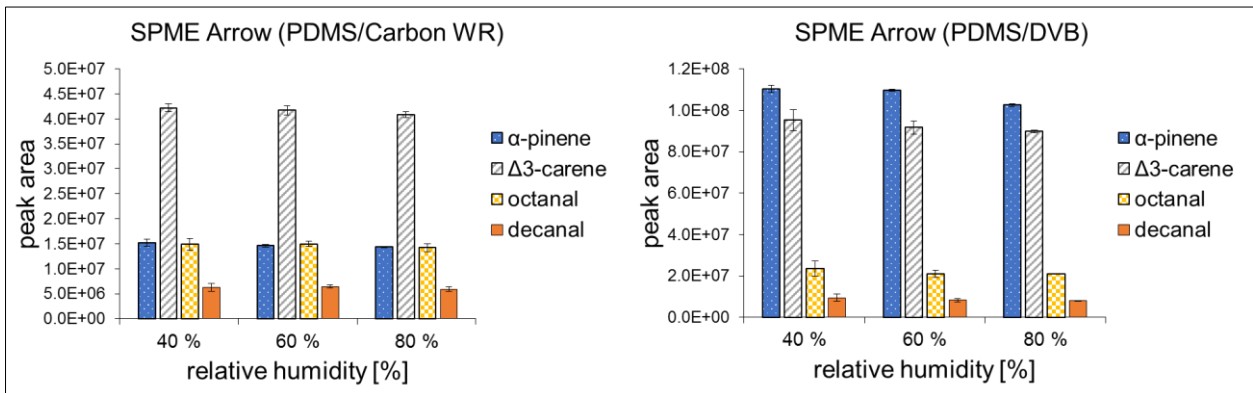

5    **Figure 2: Effect of relative humidity (%) on the extraction efficiencies obtained with SPME ARROW using with PDMS/Carbon WR and PDMS/DVB sorbents.**

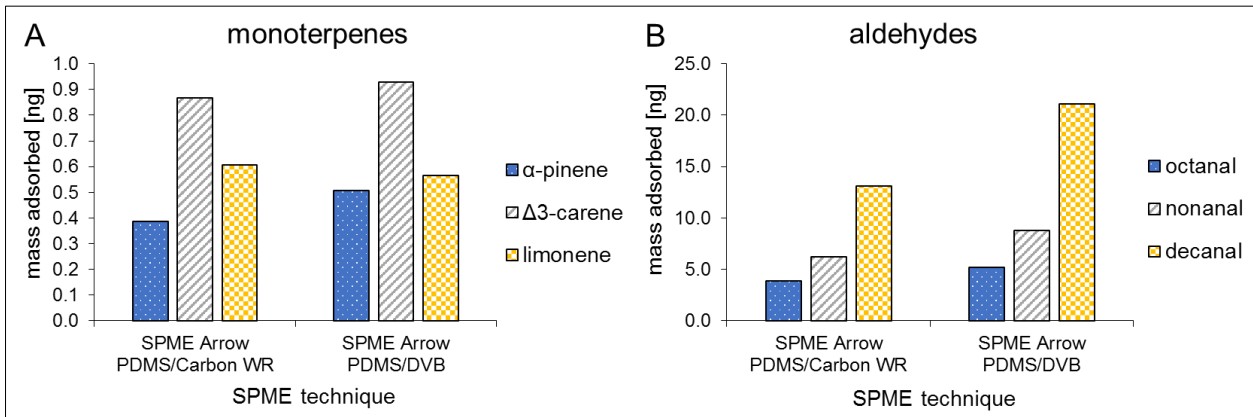

**Figure 3: Comparison between the mass of identified monoterpenes (α-pinene, Δ³-carene and limonene) and aldehydes (octanal, nonanal and decanal) collected with different SPME Arrows (PMDS/DVB and PDMS/Carbon WR) from ambient air and measured**
10    **by GC-MS.**



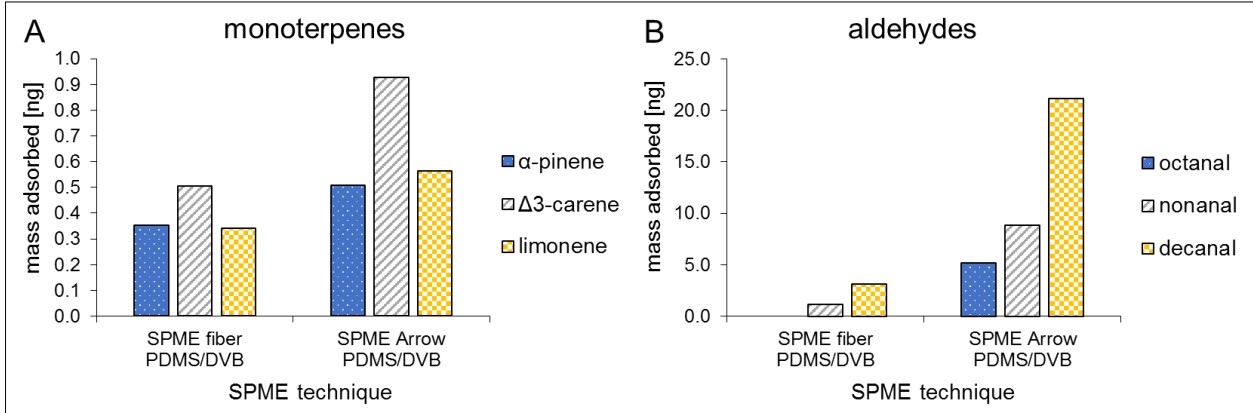

**Figure 4: Comparison between the mass of identified monoterpenes (α-pinene, Δ³-carene and limonene) and aldehydes (octanal, nonanal and decanal) collected with different PDMS/DVB SPME devices (fiber and Arrow) from ambient air and measured by GC-MS.**

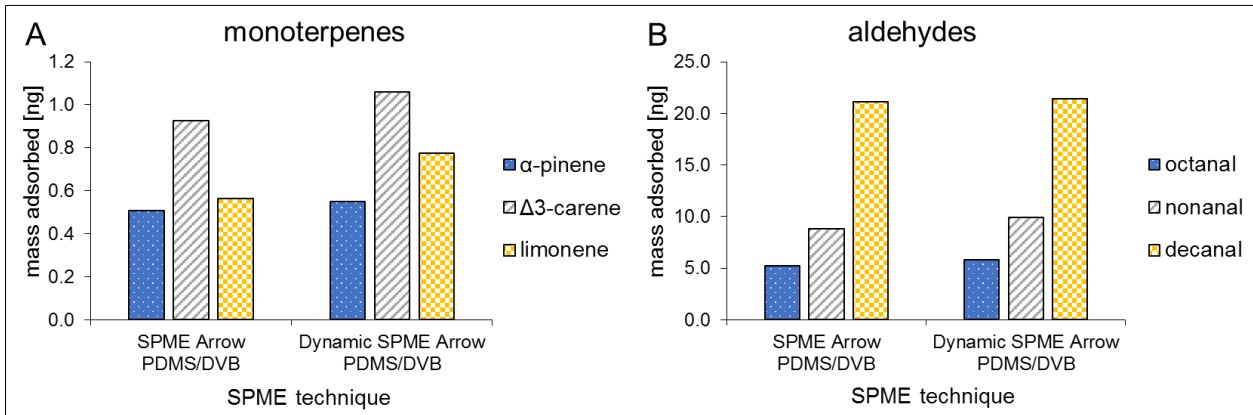

**Figure 5: Comparison between the mass of identified monoterpenes (α-pinene, Δ³-carene and limonene) and aldehydes (octanal, nonanal and decanal) collected with different PDMS/DVB SPME modes (static and dynamic) from ambient air and measured by GC-MS.**





**Figure 6: Effect of temperature (°C), relative humidity (%), photosynthetically active radiation (µmol.m⁻².s⁻¹), precipitation (mm), ozone (ppb) and particle number concentration (#.cm⁻³) on the mass of monoterpenes adsorbed on the SPME Arrows used in this study (PDMS/DVB and PDMS/Carbon WR) and measured by GC-MS.**



**Figure 7: Effect of temperature (°C), relative humidity (%), photosynthetically active radiation (μmol.m$^{-2}$.s$^{-1}$), precipitation (mm), ozone (ppb) and particle number concentration (#.cm$^{-3}$) on the mass of aldehydes adsorbed on the SPME Arrows used in this study (PDMS/DVB and PDMS/Carbon WR) and measured by GC-MS.**