# Peer review of "Field measurements of biogenic volatile organic compounds in the atmosphere using solid-phase microextraction Arrow"

_Atmospheric Measurement Techniques, 2017_

## Referee Comment (RC1) · Anonymous Referee #1 · 18 Nov 2017

**General comments**:

This paper describes the performances of SPME Arrow compared with SPME fibers for the detection of BVOCs such as monoterpenes and aldehydes. Kinetics of the extraction, the comparisons of the extraction efficiencies between SPME Arrow and SPME fibers and between coated materials, the effect of temperature and humidity on the extraction efficiency were systematically investigated in the laboratory. Then, the SPME system was tested for the field measurement of BVOCs in forest atmosphere. The dependence of BVOCs concentrations on meteorological parameters such as temperature, humidity, precipitation, PAR, ozone, and PNC was discussed. I feel that the present work was well-organized and that the paper is generally well-written.

[Figure]

But there were some parts which I did not understand. Especially, I felt that the discussions in Sec. 3.4 were vague and subjective, and were not based on statics. In the field data, the difference between SPME Arrow (PDMS/DVB) and SPME Arrow (PDMS/Carbon WR) was observed although the data were calibrated. The authors should discuss if some meteorological parameters could cause the difference or not. Therefore, I recommend this paper to be revised considering my specific comments below, before it is acceptable in Atmospheric Measurement Techniques.

**Specific comments**:
(1) Page 6, Lines 16−17: What is the sampling time? 10-min?
(2) Page 7, Line 2: Does it lead to only "underestimation"? Doesn't it happen to overestimate?
(3) Page 7, Lines 15−24: What is the sampling time and the temperature in the caribration? And which type of SPME was calibrated? In Table S1, the calibration carves for only one type of SPME are listed. Later, field data obtained from all types of SPME were quantified. Therefore, calibration carves for all types of SPME should be shown.
(4) Page 9, Line 4: Is it "the ratio between nonanal and decanal"? I think that the authors did not show the results of nonanal in the laboratory experiments.
(5) Page 9, Line 17−Page 11, Line 5: I felt that the discussions here were vague and subjective, and were not based on statics.
What is the reason of the difference between SPME Arrow (PDMS/DVB) and SPME Arrow (PDMS/Carbon WR) in Figs. 6 and 7?
(6) Page 10, Lines 6−7: What is the reason of the anti-correlation between the measured monoterpenes and PAR?
(7) Page 10, Lines 13−14: This sentence is inconsistent with the previous two sentences. Explain the reason more carefully.
(8) Page 10, Lines 21−22: Is negative effect of RH and precipitation on the amount of aldehydes surely "evident"? Discuss more carefully.

(9) Page 10, Line 26: Insert "not": . . ..any correlation was "not" observed. . . Am I right?

(10) Page 14, Figs. 1 and 2: In the results of SPME Arrow (PDMS/DVB), the values of peak area of $\alpha$-pinene are larger than those of $\Delta^3$-carene. According to kinetic data shown in Figs. S2 and S3, the values of peak area of $\alpha$-pinene are smaller than those of $\Delta^3$-carene. Are they consistent?

How were the error bars calculated? Define them.

(11) Figs. S2−S7: How were the error bars calculated? Define them.

(12) Fig. S11: The data of meteorological parameters seems to be different from those in Figs. 6 and 7. Is it just a careless mistake?

The title of the left axis should be "ratio". It should not be "mass adsorbed (ng)".
* * *

---

## Referee Comment (RC2) · Anonymous Referee #2 · 21 Nov 2017

The study by L. M. F. Barreira et. al. describes a solid-phase microextracion (SPME) Arrow, intended to quantify atmospheric relevant monoterpenes and aldehydes. The sampling capacity is compared with the conventional SPME fibers and the results indicate considerable improvements for monoterenes and aldehydes. In addition, different sorbent materials were tested and characterized under both laboratory and field conditions. The manuscript is very well written and deserves publication but several points that have to be addressed in a technical paper to warrant publication in AMT.

General comments

- I would have liked to see a detailed characterization of the temperature effect as this

parameter is the main regulating force of moterepene mixing ratios in a forested environment. The authors indicate considerable differences for a-pinene and D3-carene but I am still a bit confused on how one can deal with these effects under field conditions. I would assume that a reasonable approach would be to characterize the extraction efficiency along a wider range of temperatures and apply the respective correction under field conditions.

- It would have been valuable if the authors could demonstrate the advantages of increased sampling capacity as a function of detection limits.

- The amount of field data extremely low (15 points) and the assumptions on correlation (or not) with meteorological and environmental parameters should be more carefully approached and discussed. This is of particular importance especially when considering the title of the manuscript that at this stage could be misleading.

Specific comments

P1L19. If there is a technical possibility to additionally evaluate the effects of ozone under relevant atmospheric conditions it would have been a valuable addition to this study.

P2L26-L27. I don't think that the conventional GC-MS techniques are "laborious, expensive and prone to contamination". GC-MS is the most widely used technique for quantification of ambient monoterpenes, with no interferences on temperature or wind speed as demonstrated for SPME. Therefore, I would suggest removing this sentence completely.

P4L16. I was wondering if the sampling time could be reduced and if you could explain the reasons that you have chosen this approach. 45minutes is long time for sampling ambient mixing ratios that can dramatically change in such time frame.

P4L27. Have you tried to develop a method under SIM mode? It is commonly known that the sensitivity is much reduced when using a mass scan.

P5 and P6. I would suggest bringing supplementary figures S2, S3 and S4 into the main paper.

P7L13-14. If the extraction efficiency is reduced at higher temperatures (where we expect higher emissions) the final results will be heavily influenced. Please elaborate in detail.

P7L16. A plot (rather than a table) would have been more useful.

Fig. 4. How many samples were used? Please add errorbars.

Section 3.4: The effects of meteorological parameters should be addressed in a more comprehensive and detailed manner. Some conclusions (e.g. P9L29-31, P10L3-4, P10L7-15) are drawn very easily and without strong evidence driven by data. I would suggest to completely re-write this section, using softer language and presenting the available data in a different manner. Maybe xy plots (where x is the SPME arrow PDMS/DVB and y the PDMS/Carbon WR) colored by a 3rd dimension which would be the respective meteorological parameter, would depict better both the differences between the materials and the impact of the parameters. It is however understandable that eg. temperature did not vary significantly during the sampling period but I would have liked to understand why the two materials match on the 12.8 and have 100% difference on the 14.08.

P10L11. While this may be true, no real evidence is provided that PNC was increased due to a particle formation event of not due to some transport. Please provide some evidence or revise the text accordingly.

P10L25-26. As above.

P10L31. The authors correctly recognize but only briefly discuss the limitations of their dataset.

P11L20. It is a bit strange to see a technical paper entitled “Field measurements. . .” to use as final statement the fact that “more studies are needed to develop a proper

calibration method for field measurements”. It denotes that the current manuscript did not sufficiently fulfill its purpose. Please revise.

---

## Author Response (AR1)

Answers to reviewer's comments

Reviewer 1.

(1) Page 6, Lines 16−17: What is the sampling time? 10-min?

A: The sampling time was indeed 10 minutes. The information was added to the text.

Text was modified on page 6, line 23: Static and dynamic collections were performed during 10 minutes.

(2) Page 7, Line 2: Does it lead to only "underestimation"? Doesn't it happen to overestimate?

A: In the particular case of α-pinene, an underestimation will be observed in comparison with the other studied monoterpenes that are less volatile. However, it is true that an overestimation will be most likely observed if other monoterpenes with higher volatility than α-pinene are measured.

Text was modified on page 7, line 10: An underestimation of the measured amounts of a compound with higher volatility or an overestimation relatively to the most volatile compounds are then expected when quantifying monoterpenes under field conditions where temperature changes can be significant.

(3) Page 7, Lines 15−24: What is the sampling time and the temperature in the calibration? And which type of SPME was calibrated? In Table S1, the calibration carves for only one type of SPME are listed. Later, field data obtained from all types of SPME were quantified. Therefore, calibration carves for all types of SPME should be shown.

A: The calibration of the instrument response during field measurements was performed by using liquid standard solutions at different concentrations. The SPME systems were not calibrated to allow their comparison.

Text was modified on page 7, line 30: The calibration of instrument response was performed for the field measurement of monoterpenes (α-pinene, $\Delta^3$-carene and limonene) and aldehydes (octanal, nonanal and decanal), to estimate the mass adsorbed on the coating materials of the different SPME-based systems (Table S1). Liquid standard solutions at different concentrations were used for this purpose.
Text was modified in page 8, line 16: However, when the calibration of analytes response was performed with standard solutions (but not the calibration of SPME collection), $\Delta^3$-carene and limonene levels increased relatively to α-pinene (Fig. 6A).

(4) Page 9, Line 4: Is it "the ratio between nonanal and decanal"? I think that the authors did not show the results of nonanal in the laboratory experiments.

A: The first mentioned ratio is the one obtained during field measurements between decanal and nonanal. Octanal was not measured with SPME fiber. However, during laboratory experiments, the tests were performed for octanal and decanal and the second mentioned ratio is the one obtained between decanal and octanal. Even though these ratios could probably still be compared because analytes belong to the same homologous series, the sentence was eliminated since additional tests would be required to confirm this possibility.

The sentence was removed.

(5) Page 9, Line 17−Page 11, Line 5: I felt that the discussions here were vague and subjective, and were not based on statics. What is the reason of the difference between SPME Arrow (PDMS/DVB) and SPME Arrow (PDMS/Carbon WR) in Figs. 6 and 7?

A: the whole section 3.4 was completely rewritten to emphasize that the purpose of the section was to evaluate preliminarily the effect of atmospheric parameters on SPME sampling. In Figs 6 and 7, the trends between SPME Arrow (PDMS/DVB) and SPME Arrow (PDMS/Carbon WR) were similar. The differences in the extracted masses are likely related with a combination of several factors, including the distinct extraction efficiencies, competitive adsorption and different effects of atmospheric parameters (e.g. temperature and relative humidity).

[revised manuscript text omitted]

(6) Page 10, Lines 6−7: What is the reason of the anti-correlation between the measured monoterpenes and PAR?

A: At summer time, the main factor determining monoterpene emissions at the boreal forest is temperature due to the temperature-dependent residence in specific storage structures located internal or external to

the leaf. However, a PAR effect on monoterpene emissions was also found to be important (e.g. at spring-time when temperature is still low). In our study, the anti-correlation between PAR and monoterpenes is most likely not related with lower emissions from vegetation. Higher PAR is usually observed in the beginning of afternoon, which usually coincides with the period of the day with higher atmospheric reactivity. This reactivity is most likely the main reason why a correlation between PAR and monoterpenes is not observed. A discussion was added to the text.

Text was modified in page 10, line 27: Due to the constancy of temperature during the sampling campaign, ozone and PAR were also expected to affect significantly the measured amounts of monoterpenes. Indeed, some anti-correlation was found between the measured monoterpenes and these parameters. This result is likely to reflect the increased photooxidation during periods of the day when PAR is high, since the effects of temperature and/or light on monoterpene emissions have been described previously (Aalto et al., 2014).

(7) Page 10, Lines 13−14: This sentence is inconsistent with the previous two sentences. Explain the reason more carefully.

A: The sentence was revised.

Text was modified in page 10, line 33: PNC also seemed to increase with the amounts of monoterpenes present in the ambient air. This result is expected, since monoterpene oxidation in the atmosphere and consequent formation of low volatile compounds have been recognized to contribute to aerosol particle formation (Laaksonen et al., 2008). However, other factors can also contribute to the increase in PNC, such as long-range transport.

(8) Page 10, Lines 21−22: Is negative effect of RH and precipitation on the amount of aldehydes surely "evident"? Discuss more carefully.

A: A more careful discussion was added to the text.

Text was modified in page 11, line 8: Relative humidity and precipitation also coincided with a burst in aldehyde atmospheric amounts, excluding a negative effect of this parameter on the SPME sampling, but seemed to anti-correlate with these parameters when aldehyde amounts were low. This observation can be a consequence of the solubility of these compounds in water at low concentrations, but additional studies are also required to confirm this hypothesis.

(9) Page 10, Line 26: Insert "not": . . ..any correlation was "not" observed. . . Am I right?

A: The sentence was corrected.

Text was modified in page 11, line 15: The effect of aldehyde amounts on PNC was not very clear, which might be a consequence of the lower atmospheric reactivity of these compounds. No correlation was

observed between atmospheric parameters and the ratio between the amounts collected on PDMS/Carbon WR and PDMS/DVB, which agrees with the non-dependences on temperature and relative humidity verified in laboratory studies.

(10) Page 14, Figs. 1 and 2: In the results of SPME Arrow (PDMS/DVB), the values of peak area of $\_$-pinene are larger than those of $\_3$-carene. According to kinetic data shown in Figs. S2 and S3, the values of peak area of $\_$-pinene are smaller than those of $\_3$-carene. Are they consistent?
How were the error bars calculated? Define them.

A: The concentrations of analytes were not changed during all experiments and it is true that α-pinene was expected to be slightly higher during SPME Arrow (PDMS/DVB) kinetic studies and during comparison between extraction efficiencies. However, considering the error bars, the amounts of α-pinene and $\Delta^3$-carene were quite similar when using a PDMS/DVB sorbent and results are still consistent with the SPME fiber coated with the same extraction material (Figs. S2, S4 and S7).
The error bars in Figs. 1 and 2 are the standard deviations from 3 repetitions. In Figs. S2-S3, three repetitions were performed at 10 minutes and the obtained RSD was used for all the calibration. The RSD will be likely higher at lower sampling times and lower for the longer sampling times. However, because most of the laboratory tests were performed at 10 minutes (except for temperature and RH studies that were performed at 20 minutes) and field measurements were performed at longer sampling times (45 minutes), the used RSD is representing a worst-case scenario in the kinetic studies.

Text was modified in page 5, line 24: For an estimation of standard deviations during kinetic studies, 3 repetitions were performed at 10 minutes, corresponding to the minimum sampling time used during all other experiments and consequently to the higher expected variation.
Text was modified in page 6, line 32: Three replicates were performed for each temperature and relative humidity studied.

(11) Figs. S2−S7: How were the error bars calculated? Define them.

A: The error bars in Figs. S2-S3 were calculated as described in the previous question. In Figs S4-S7, three repetitions were performed and the obtained standard deviations were used as error bars.

Text was modified on page 5, line 24: For an estimation of standard deviations during kinetic studies, 3 repetitions were performed at 10 minutes, corresponding to the minimum sampling time used during all other experiments and consequently to the higher expected variation.
Text was modified on page 6, line 15: Three repetitions were performed with each SPME system.
Text was modified on page 6, line 32: Three replicates were performed for each temperature and relative humidity studied.

(12) Fig. S11: The data of meteorological parameters seems to be different from those in Figs. 6 and 7. Is it just a careless mistake?
The title of the left axis should be "ratio". It should not be "mass adsorbed (ng)"..

A: There was a mistake in the x-axis (date) that was corrected. The title of the y-axis was modified to Carbon WR/DVB (ng/ng).

Text was modified on page 6, line 1 (supplement):

[Figure]

**Figure S8: Effect of temperature (°C), relative humidity (%), photosynthetically active radiation (μmol.m$^{-2}$.s$^{-1}$), precipitation (mm), ozone (ppb) and particle number concentration (#.cm$^{-3}$) on the ratio between the amounts of monoterpenes (grey line) and aldehydes (blue line) sampled with PDMS/Carbon WR and PDMS/DVB and measured by GC-MS.**

Reviewer 2.

General comments

(1) I would have liked to see a detailed characterization of the temperature effect as this C1 parameter is the main regulating force of monoterpene mixing ratios in a forested environment. The authors indicate considerable differences for a-pinene and D3-carene but I am still a bit confused on how one can deal with these effects under field conditions. I would assume that a reasonable approach would be to characterize the extraction efficiency along a wider range of temperatures and apply the respective correction under field conditions.

A: The proposed approach is one of the possibilities. However, the ideal solution would be to eliminate this effect. For that purpose, we would suggest the construction of a chamber that can be field portable and that is able to keep the temperature constant during both sampling and calibration. However, additional research is still needed to study both of these possibilities.

Text was modified on page 7, line 13: The effect of temperature on the amounts of analytes collected by SPME must consequently be assessed or avoided during quantitative field measurements.

(2) It would have been valuable if the authors could demonstrate the advantages of increased sampling capacity as a function of detection limits.

A: Indeed, it is important to develop a calibration method so that we can determine the detection limits for all used SPME systems. However, our results showed already the benefit of having increased capacity for both qualitative (e.g. we detected octanal with SPME Arrow but not with SPME fiber) and semi-quantitative measurements (e.g. errors related to the analysis of target compounds are usually higher when responses are closer to the baseline).

No modification was done to the manuscript.

(3) The amount of field data extremely low (15 points) and the assumptions on correlation (or not) with meteorological and environmental parameters should be more carefully approached and discussed. This is of particular importance especially when considering the title of the manuscript that at this stage could be misleading.

A: the whole section 3.4 has been rewritten and more carefully discussed.

[revised manuscript text omitted]

Specific comments

(4) P1L19. If there is a technical possibility to additionally evaluate the effects of ozone under relevant atmospheric conditions it would have been a valuable addition to this study.

A: A technical possibility to evaluate the effects of ozone under relevant atmospheric conditions would be to generate different oxidants in a chamber/flow tube and perform measurements, maintaining all other conditions constant. This approach would be very challenging, since adding oxidants to the chamber will also decrease the analytes concentration. There are also some possibilities that could possibly be used with our method to avoid the effect of ozone and other oxidants on the measured amounts of VOCs (e.g. the use of an ozone scrubber or a heated stainless steel tube connected to our sampling device). These studies must be performed further.

Text was modified on page 10, line 31: Nonetheless, on-fiber oxidation might occur during SPME collection. For that reason, the effect of oxidants must be assessed further by performing complementary laboratory experiments under controlled conditions.
Text was modified on page 11, line 14: On-fiber oxidation studies are also still required for aldehydes.

(5) P2L26-L27. I don't think that the conventional GC-MS techniques are "laborious, expensive and prone to contamination". GC-MS is the most widely used technique for quantification of ambient monoterpenes, with no interferences on temperature or wind speed as demonstrated for SPME. Therefore, I would suggest removing this sentence completely.

A: The sentence was removed and the main advantage of SPME compared to TD-GC-MS was referred.

Text was modified on page 2, line 24: Alternatively, monoterpenes have been successfully sampled on tubes packed with an adsorbent material (such as Tenax TA/Carbopack-B), and subsequently desorbed into a thermal desorption-gas chromatograph-mass spectrometer for off-line or on-line analysis (e.g. Haapanala et al., 2012; Hakola et al., 2012; Wong et al., 2013). The main limitation of this method is the requirement of sophisticated instrumentation that is less convenient for field measurements (e.g. thermal desorption unit and cryofocusing). Solid-phase microextraction (SPME) has also been used for the collection of monoterpenes (e.g. Yassaa et al., 2010; Zini et al., 2001). This technique combines sampling and pre-concentration of analytes in a single step and allows for direct thermal desorption into a heated gas chromatograph injection port (Koziel et al., 1999).

(6) P4L16. I was wondering if the sampling time could be reduced and if you could explain the reasons that you have chosen this approach. 45minutes is long time for sampling ambient mixing ratios that can dramatically change in such time frame.

A: The sampling time could likely be reduced, especially when using SPME Arrow. The main reasons for using a 45 minutes sampling time were to sample detectable amounts of target analytes with all SPME systems and to reduce the significance of errors associated with the time lag between sampling and injection (even though sample modification was not expected to be significant due to the retraction of extraction materials under the SPME needle and the fact that SPME syringes were closed with a cap).

Text was modified on page 4, line 19: Samples were collected in static mode for 45 minutes, to sample detectable amounts of target analytes with all SPME systems and to reduce the significance of errors associated with the time lag between sampling and injection.

(7) P4L27. Have you tried to develop a method under SIM mode? It is commonly known that the sensitivity is much reduced when using a mass scan.

A: A SIM mode increases the sensitivity of the method. However, at this stage of development, we would like to know what type of compounds can be identified and a scan mode was then chosen for that purpose. This approach allowed to study compounds that were not considered during the laboratory tests (limonene and nonanal).

Any modification was done in the manuscript.

(8) P5 and P6. I would suggest bringing supplementary figures S2, S3 and S4 into the main paper.

A: The figures were brought to the main paper.

Text was modified on page 17, line 1: Figs. S2, S3 and S4 modified to Figs. 1, 2 and 3.

(9) P7L13-14. If the extraction efficiency is reduced at higher temperatures (where we expect higher emissions) the final results will be heavily influenced. Please elaborate in detail.

A: The sentence was corrected.

Text was modified on page 7, line 13: The effect of temperature on the amounts of analytes collected by SPME must consequently be assessed or avoided during quantitative field measurements.
Text was modified on page 7, line 25: Even though the effect of relative humidity was negligible in our study, this parameter might greatly influence the SPME collection when using other sorbents or when sampling other analytes.

(10) P7L16. A plot (rather than a table) would have been more useful.

A: In order to compact the information obtained during calibration studies, results were presented as a table. All relevant parameters were carefully provided in the table and in the text.

No modification was done to the manuscript.

(11) Fig. 4. How many samples were used? Please add errorbars.

A: The results are the average amounts measured during all the campaign. This information was added to the caption. Repetitions were not performed during field measurements, since a satisfactory repeatability was obtained during laboratory measurements and 4 different systems were compared simultaneously in the field (which would require 12 systems for 3 repetitions and a long time to finalize the measurements of each experiment). For that reason, error bars were only added to the laboratory experiments. The detailed results during all sampling campaign are represented in Fig. S8 (Fig. S5 in the new version of the manuscript) and Table S2, which show the consistent improvement in extraction efficiency when using SPME Arrow comparatively to SPME fiber during all sampling period.

Text was modified on page 23, line 4: Figure 7: Comparison between the average mass of identified monoterpenes ($\alpha$-pinene, $\Delta^3$-carene and limonene) and aldehydes (octanal, nonanal and decanal) collected with different PDMS/DVB SPME devices (fiber and Arrow) from ambient air and measured by GC-MS.

(12) Section 3.4: The effects of meteorological parameters should be addressed in a more comprehensive and detailed manner. Some conclusions (e.g. P9L29-31, P10L3-4, P10L7-15) are drawn very easily and without strong evidence driven by data. I would suggest to completely re-write this section, using softer language and presenting the available data in a different manner. Maybe xy plots (where x is the SPME arrow PDMS/DVB and y the PDMS/Carbon WR) colored by a 3rd dimension which would be the respective meteorological parameter, would depict better both the differences between the materials and the impact of the parameters. It is however understandable that eg. temperature did not vary significantly during the sampling period but I would have liked to understand why the two materials match on the 12.8 and have 100% difference on the 14.08.

A: All section 3.4 was completely rewritten to emphasize that the main purpose of the section was to evaluate preliminarily the effect of atmospheric parameters on the SPME sampling. In Figs 6 and 7, the trends between SPME Arrow (PDMS/DVB) and SPME Arrow (PDMS/Carbon WR) were similar. The differences in the extracted masses are likely related with a combination of several factors, including the distinct extraction efficiencies, competitive adsorption and different effects of atmospheric parameters (e.g. temperature and relative humidity).

Corrected as described above.

(13) P10L11. While this may be true, no real evidence is provided that PNC was increased due to a particle formation event of not due to some transport. Please provide some evidence or revise the text accordingly. P10L25-26. As above.

A: The sentences were revised.

Text was modified on page 10, line 33: PNC also seemed to increase with the amounts of monoterpenes present in the ambient air. This result is expected, since monoterpene oxidation in the atmosphere and consequent formation of low volatile compounds have been recognized to contribute to aerosol particle formation (Laaksonen et al., 2008). However, other factors can also contribute to the increase in PNC, such as long-range transport.
Text was modified on page 11, line 15: The effect of aldehyde amounts on PNC was not very clear, which might be a consequence of the lower atmospheric reactivity of these compounds.

(14) P10L31. The authors correctly recognize but only briefly discuss the limitations of their dataset.

A: The limitations of our data set were clarified.

Text was modified on page 11, line 19: Even though the effects of atmospheric parameters on the SPME sampling were preliminarily evaluated with our method under atmospheric relevant conditions, longer data sets and quantitative data are needed to estimate accurately the correlation of these parameters with BVOC mixing ratios.
Text was modified on page 13, line 26: Longer data sets are also required to study in more detail the effects of atmospheric parameters on the SPME sampling under atmospheric relevant conditions.

(15) P11L20. It is a bit strange to see a technical paper entitled "Field measurements. . ." to use as final statement the fact that "more studies are needed to develop a proper calibration method for field measurements". It denotes that the current manuscript did
not sufficiently fulfill its purpose. Please revise.

A: The sentence was revised.

[revised manuscript text omitted]